*Experimental Results* (2021), 2, e15, 1–10

CAMBRIDGE
UNIVERSITY PRESS

**LIFE SCIENCE AND BIOMEDICINE**

NOVEL-RESULT

# Weather variability and transmissibility of COVID-19: a time series analysis based on effective reproductive number

Xiaohan Si[1] , Hilary Bambrick[1], Yuzhou Zhang[1], Jian Cheng[1], Hannah McClymont[1] , Michael B. Bonsall[2] and Wenbiao Hu[1,*] 

[1]School of Public Health and Social Work, Queensland University of Technology, Brisbane, 4059, Queensland, Australia, and
[2]Mathematical Ecology Research Group, Department of Zoology, University of Oxford, Oxford, OX1 3SZ, UK
*Corresponding author: E-mail: w2.hu@qut.edu.au

(Received 25 January 2021; Revised 09 February 2021; Accepted 15 February 2021)

**Abstract**

COVID-19 is causing a significant burden on medical and healthcare resources globally due to high numbers of hospitalisations and deaths recorded as the pandemic continues. This research aims to assess the effects of climate factors (i.e., daily average temperature and average relative humidity) on effective reproductive number of COVID-19 outbreak in Wuhan, China during the early stage of the outbreak. Our research showed that effective reproductive number of COVID-19 will increase by 7.6% (95% Confidence Interval: 5.4% ~ 9.8%) per 1°C drop in mean temperature at prior moving average of 0–8 days lag in Wuhan, China. Our results indicate temperature was negatively associated with COVID-19 transmissibility during early stages of the outbreak in Wuhan, suggesting temperature is likely to effect COVID-19 transmission. These results suggest increased precautions should be taken in the colder seasons to reduce COVID-19 transmission in the future, based on past success in controlling the pandemic in Wuhan, China.

**Keywords:** weather factors; COVID-19; effective reproductive number; time series regression model

## 1. Introduction

COVID-19 is a widespread global pandemic caused by SARS-CoV-2 coronavirus causing significant socio-economic impact. As of 8[th] February 2021, over the duration of the pandemic over 106 million confirmed cases and 2 million deaths have been reported in over 200 countries, areas or territories (Johns Hopkins University, 2020). Previous studies show that cold and dry weather may positively influence coronavirus survival time and the transmission rate of upper respiratory tract coronavirus infections (Chan et al., 2011; Van Doremalen et al., 2013). However, the role of meteorological effect on the spread of COVID-19 is still controversial (Demongeot et al., 2020; Qi et al., 2020; Tosepu et al., 2020). A recent review on 23 studies about weather and COVID-19 showed that temperature and humidity can contribute to increased transmission of COVID-19, particularly in winter conditions which is a conductive environment for virus survival (McClymont & Hu, 2021). For example, a study on the growth of new cases in tropical and temperate regions showed that temperature has a positive association with the number of daily new cases (Chennakesavulu & Reddy, 2020). Another study on multiple cities in China indicated that temperature and absolute humidity are negatively associated with COVID-19 daily

new cases (Liu et al., 2020). However, these studies only used confirmed case number as response variable, rather than transmissibility, to investigate the climate effect in COVID-19 transmission.

Effective reproductive number ($R_{eff}$) is a robust model-based indicator of transmissibility of COVID-19, which can reflect the real-time transmissibility of COVID-19 through an outbreak. The research aims to assess the effects of climate factors (i.e. daily average temperature and average relative humidity) on $R_{eff}$ of COVID-19 based on local cases in Wuhan, China. The $R_{eff}$ we used in this study considered the effect from public health interventions, providing more information and reducing confounding factors compared with daily cases. Also, compared with research only reporting the basic reproductive number ($R_0$), $R_{eff}$ can also provide greater accuracy in estimating transmissibility as it does not need to meet the model assumption that the virus is freely transmitted with no intervention (Nishiura & Chowell, 2009).

## 2. Data collection and statistical methods

Wuhan city was chosen as our study site due to strict lockdown measures which were implemented from 23[rd] January 2020. During this time, all cases included in our study could be considered as locally transmitted without any imported cases during this period. This environment is ideal for modelling the COVID-19 transmission dynamic via effective reproductive number based on Susceptible-Infected-Removal (SIR) model from Bayesian estimation.

Daily number of confirmed COVID-19 cases reported between 14[th] January 2020 and 17[th] March 2020 were obtained from the JHU coronavirus resource center (Dong et al., 2020). Daily data on temperature and humidity through the same period were obtained from National Climatic Data Center, US Department of Commerce (https://www.ncdc.noaa.gov).

We used Bayesian Estimation theory to estimate the $R_{eff}$, and a 10-days averaging window was applied to reduce the impact from stochastic events (e.g., population migration during Chinese New Year). The method was used to estimate daily parameter of SIR model and calculate daily $R_{eff}$ (Cori et al., 2013; Forsberg White & Pagano, 2008). The mean and standard deviation of the incubation period used in this study are 4.7 days and 2.9 days respectively, estimated by Nishiura et al. (2020). Besides a 10 day time window was assigned in estimation to improve the estimation accuracy via *EpiEstim* (Cori et al., 2013). To control for alterations to case reporting criteria on 12[th] February, only laboratory confirmed cases were included, which for this date was 1,072 cases, issued by Health Commission of Hubei Province (Health Commission of Hubei Province, 2020).

Time series generalized linear model (GLM) and generalized additive model (GAM) with Gamma distribution and logarithm link function were used to assess the relationship between meteorological factors and $R_{eff}$. The GLM can be defined as:

$$log(Y_t) = \beta_0 + \beta_1 TEMP_t + \beta_2 RH_t \tag{1}$$

where $Y_t$ denotes the estimated $R_{eff}$ at time t during the outbreak. $\beta_0, \beta_1$ and $\beta_2$ represents the model intercept and coefficients of independent variables under lag effect on average temperature (TEMP) and average relative humidity (RH), respectively. We used cross-correlation function (CCF) to evaluate the lag effect at different days between TEMP, RH and COVID-19 transmission. Based on the assessment of CCF, we used lag 0–8 days on TEMP and lag 0–3 days on RH in our model.

The GAM can be defined as:

$$log(Y_t) = \beta_0 + s(TEMP_T, df = 4) + s(RH_t, df = 4) \tag{2}$$

where $s(\cdot)$ is the thin plate regression spline function for smoothing, which are average temperature and average relative humidity at time t; df is the degrees of freedom; $Y_t$ and $\beta_0$ represents the same term as in model (1).

Sensitivity analysis on the degree of freedom (df) of smoothing spline showed that the model has better performance in generalized cross-validation (GCV) when df = 4 (k-index = 0.82, p = 0.12) compared with the model when df = 3 (k-index = 0.83, p = 0.07) and df = 2 (k-index = 0.83, p = 0.04).

## 3. Results

Figure 1 showed that the real-time changes of $R_{eff}$ during the outbreak had a gradual downward trend until the end of February.

Table 1 showed that a moving average of lag 0–8 days temperature after adjusted with RH was associated with daily $R_{eff}$ in Wuhan city (Relative Risk (RR): 0.924, 95% Confidence Interval (CI): 0.902–0.946). However, there was no association between RH and $R_{eff}$.

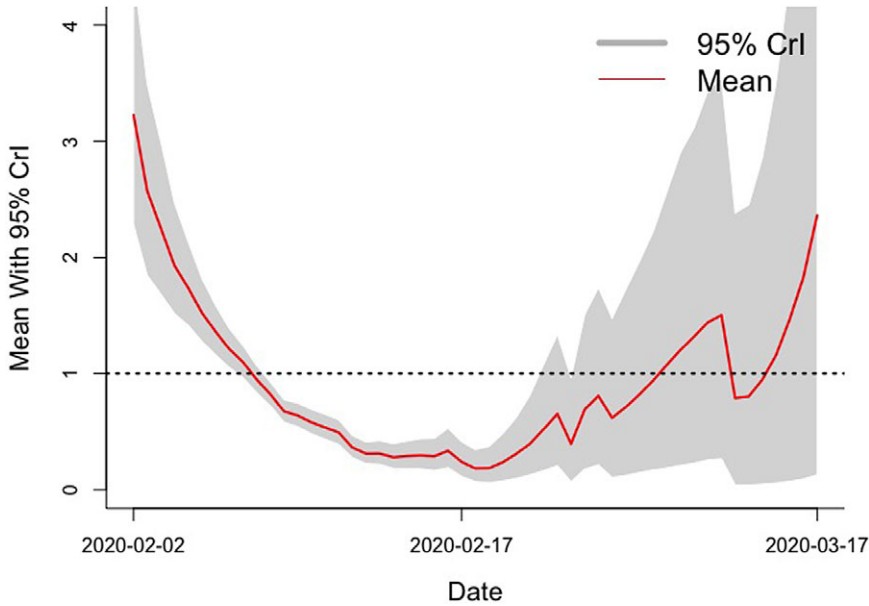

**Figure 1.** Daily $R_{eff}$ curve in Wuhan, China

**Table 1.** Relative risks of $R_{eff}$ from time series generalized linear model

| Variables | Relative Risk | 95% CI | p-value |
|---|---|---|---|
| TEMP (°C) | 0.924 | (0.902, 0.946) | <0.001 |
| RH(%) | 1.007 | (0.906, 1.024) | 0.394 |

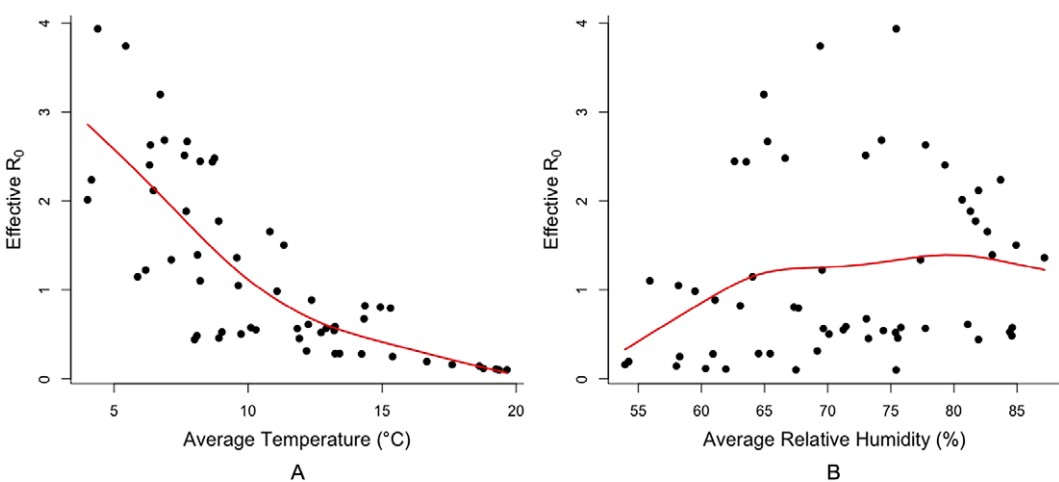

**Figure 2.** The smoothing spline with 4 degrees of freedom between daily $R_{eff}$ and TEMP (panel A); daily $R_{eff}$ and RH (panel B).

As for GAM, Figure 2 showed the scatter plots with smoothing spline with 4 degrees of freedom from the model between daily $R_{eff}$, TEMP and RH respectively. Figure 2 suggested that TEMP has an approximate linear relation with $R_{eff}$, but average relative humidity shows more nonlinear variation in the relationship.

## 4. Conclusions

Our research based on the $R_{eff}$ suggested that temperature has a significant negative association with COVID-19 transmission. This result was similar to results reported in some studies for the association between temperature and COVID-19 case numbers (Tosepu et al., 2020; Zhu & Xie, 2020). However, relative humidity was not found to contribute significantly in explaining variation in transmissibility of COVID-19. Our research filled the gap in illustrating the weather effect on COVID-19 transmission rate via $R_{eff}$ with strong public health interventions.

Our result gives a higher variation range of $R_{eff}$ by 1°C changes on temperature (5.4% ~9.8%) compared with previous studies (2%~4% or no effects) which excluded intervention effects and used reproductive number looking at virus transmission (Sahafizadeh & Sartoli, 2020; Wang et al., 2020). One possible explanation for this difference is the choice of time period: a study assessed a very short time period with no public health intervention in China (19 January–23 January) and U.S. (15 March–6 April), and all data were from the period prior to stay-at-home orders being fully implemented (Wang et al., 2020). Furthermore, a large number of cities in different locations were included in some studies to estimate transmissibility of COVID-19 contemporaneously but without a single region or area having sufficient epidemic duration for a more robust analysis (Baker et al., 2020; Liu et al., 2020; Qi et al., 2020; Zhu & Xie, 2020). Our two-month study period is long enough to be considered as a full epidemic outbreak, given the lag for incubation period of up to 14 days for transmission of this disease. Another focus which might cause such variation is the potential nonlinear relation between $R_{eff}$ and studying meteorological variable. Future studies are required to explore these complex nonlinear and interactive effects on virus transmission risk.

Previous research showed absolute or relative humidity had positive or negative influence on COVID-19 transmission (Guo et al., 2020; McClymont & Hu, 2021; Park et al., 2019). These studies reported marginal correlations between humidity and COVID-19 cases number. Further potential caveats is the classification of imported versus local cases, which might contribute to the wide variation in transmission between different groups and potential for super spreaders (Baker et al., 2020; Liu et al., 2020; Qi et al., 2020; Yao et al., 2020). The misclassification between imported cases and local transmitted cases might lead to the overestimation of transmissibility of COVID-19, especially in those countries or regions where imported cases are the cases majority.

Two other studies used time series Auto Regressive Integrated Moving Average (ARIMA) model and machine learning as new approaches to assess the association between weather and COVID-19 cases (Malki, Atlam, Ewis, et al., 2020; Malki, Atlam, Hassanien, et al., 2020). However, ARIMA model requires a relatively long time period and stationarity of time series data (Chintalapudi et al., 2020). In future research, seasonal ARIMA modelling can be used to predict the trend of COVID-19 transmission, with potential socio-environmental factors.

A possible limitation of this study is that we used the case notification date rather than onset date due to data availability which may have reduced the strength and/or accuracy of the estimated relationship between temperature and $R_{eff}$. UV radiation and air pollution were not included in our study. UV radiation is associated with COVID-19 transmission as reported in several studies (Cadnum et al., 2020; Hamzavi et al., 2020). However, UV radiation will only be relevant to outdoor human activities and subsequent transmission, due to strict lockdowns, Wuhan city banned all outdoor activities for the entirety of the study period (Pun et al., 2020). Moreover, air pollutant significantly reduced during the study period in Wuhan while the city was in lockdown (Lian et al., 2020). Under these conditions, it is reasonable to believe UV radiation and air pollutant could have a limited effect on COVID-19 transmission.

In conclusion, our study suggested that temperature changes have an effect on the transmissibility of COVID-19, with transmission increasing as temperature declines. However, further research is required to assess the complex relationship at global, regional and local levels, and develop a spatiotemporal weather-based early warning system for COVID-19.

**Acknowledgements.** We acknowledge the work and contribution of all workers in National Health Commission of China, Health Commission of Hubei Province and Johns Hopkins University Coronavirus Resource Center on collecting and publishing data of COVID-19 outbreak, and of the National Climatic Data Center, US Department of Commerce on climate data publication. We gratefully acknowledge the efforts of healthcare workers in Wuhan, and globally, caring for patients infected with SARS-CoV-2.

**Author contributions.** WH conceived and designed the study. XS and YZ conducted data gathering. XS performed statistical analyses and drafted the paper. All authors interpreted the results and revised the paper.

**Funding information.** This research received no specific grant from any funding agency, commercial or not-for-profit sectors.

**Conflicts of interest.** All authors declare none.

**Availability of data.** All data used in this article are publicly available.

Daily counts of reported confirmed COVID-19 cases, recoveries and deaths for each country are obtained from the JHU coronavirus resource center [1, 18] (publicly available at https://github.com/CSSEGISandData/COVID-19).

Weather data is available from National Climatic Data Center, US Department of Commerce (https://www.ncdc.noaa.gov).

**Code availability.** Using *R* 4.0.0 with package *EipEstim 2.2* and *mgcv 1.8* for modelling.

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

# Peer Reviews

**Reviewing editor:** Dr. Michael Nevels

University of St Andrews, Biomolecular Sciences Building, Fife, United Kingdom of Great Britain and Northern Ireland, KY16 9ST

This article has been accepted because it is deemed to be scientifically sound, has the correct controls, has appropriate methodology and is statistically valid, and has been sent for additional statistical evaluation and met required revisions.

doi:10.1017/exp.2021.4.pr1

## Review 1: Weather variability and COVID-19 transmission rates: a time series analysis based on effective reproductive number

**Reviewer:** Dr. Yiqun Ma 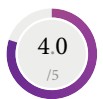

Yale University School of Public Health, Environmental Health Sciences, 60 College Street, New Haven, Connecticut, United States, 06520-8034

Date of review: 31 January 2021

**Conflict of interest statement.** Reviewer declares none.

*Comments to the Author:* This study investigated the effects of air temperature and relative humidity on SARS-CoV-2 transmission in Wuhan, China. Great improvements have been made in this manuscript, such as the brief review of existing studies, the sensitivity analyses by testing different degrees of freedom, and more justification in the discussion part. There are several minor issues needed to be addressed before publication:

1. The authors used lag of 5 days for temperature and 8 days for RH in the statistical model. Explanations on how these lag durations were chosen is necessary. In addition, the selection of lag period could influence the final estimates. Sensitivity analyses using different lags are needed.

2. Line 94-96, it is good to perform these sensitivity analyses, but please display model statistics to support this statement.

3. In the description of the GAM, it is unclear which type of smoothing basis was used (i.e., bs=? in the s(.))?

4. There are still several typos in the text (e.g., line 49, "tete"), please check again in the revision.

## Score Card
### Presentation

**4.0** /5

| | |
|---|---|
| Is the article written in clear and proper English? (30%) | 4/5 |
| Is the data presented in the most useful manner? (40%) | 4/5 |
| Does the paper cite relevant and related articles appropriately? (30%) | 4/5 |

## Context

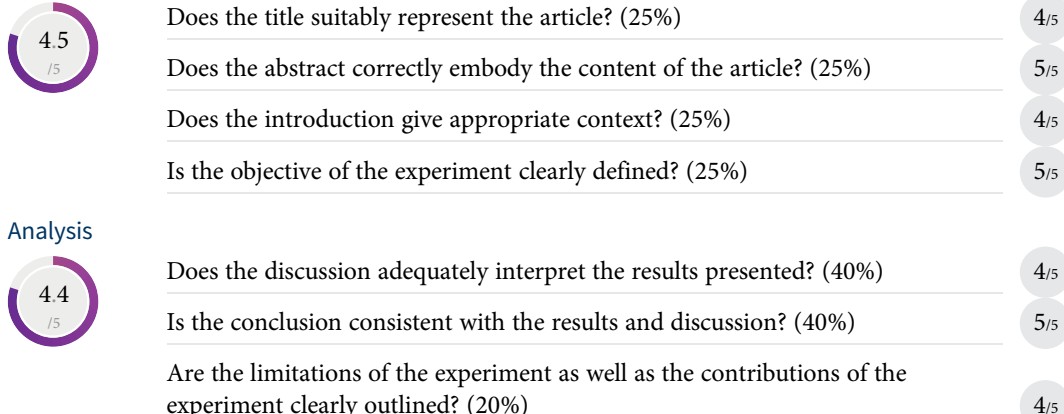

| | |
|---|---|
| Does the title suitably represent the article? (25%) | 4/5 |
| Does the abstract correctly embody the content of the article? (25%) | 5/5 |
| Does the introduction give appropriate context? (25%) | 4/5 |
| Is the objective of the experiment clearly defined? (25%) | 5/5 |

4.5 /5

## Analysis

| | |
|---|---|
| Does the discussion adequately interpret the results presented? (40%) | 4/5 |
| Is the conclusion consistent with the results and discussion? (40%) | 5/5 |
| Are the limitations of the experiment as well as the contributions of the experiment clearly outlined? (20%) | 4/5 |

4.4 /5

doi:10.1017/exp.2021.4.pr2

# Review 2: Weather variability and COVID-19 transmission rates: a time series analysis based on effective reproductive number

**Reviewer:** Dr. El-Sayed Atlam ⓘ

Date of review: 02 February 2021

**Conflict of interest statement.** The paper has no Conflicts of Interest

*Comments to the Author:* • The manuscript can be checked by native English speaker
 • The main contribution and originality should be explained in more detail, is it the use of Bayesian Estimation? or the found associations?
 • Discussion of related work in COVID-19 should be expanded with more recent work, this will better situate this paper in the context of the journal
 • This paper has quite a few issues to check and perhaps to correct. But one major issue is to have and to support reproducibility of this work, to have comprehensive evaluation and applications carried out, from this work here.
 • The wider interpretation of the results can be added for the better understanding of the analysis.
 • A comparison with other methods should be explained with more detail to show the effectiveness of using the new approach== for example with the following two references:
 1- Association between Weather Data and COVID-19 Pandemic Predicting Mortality Rate: Machine Learning Approaches, Zohair Malki, El-Sayed Atlama, Aboul Ella Hassanien, Guesh Dagnew, Mostafa A. Elhosseini and Ibrahim Gad, Journal of Chaos, Solitons &Fractals, Vol. 138,110137,2020.
 2- ARIMA Models for Predicting the End of COVID-19 Pandemic and the Risk of a Second Rebound, Zohair Malki, El-Sayed Atlam, Ashraf Ewis, Guesh Dagnew, Ahmad Reda Alzighaibi, ELmarhomy Ghada, Mostaf A. Elhosseini, Aboul Ella Hassanien, Ibrahim Gad., May 27,2020, Journal of Neural Computing and Applications, May 272020-Accepted Oct.8th,2020.
 oMany references are old and recent references should be appended for good comparison to related works.

---

## Score Card

### Presentation

|  |  |  |
|---|---|---|
| **3.0** /5 | Is the article written in clear and proper English? (30%) | 3/5 |
|  | Is the data presented in the most useful manner? (40%) | 3/5 |
|  | Does the paper cite relevant and related articles appropriately? (30%) | 3/5 |

### Context

|  |  |  |
|---|---|---|
| **2.8** /5 | Does the title suitably represent the article? (25%) | 3/5 |
|  | Does the abstract correctly embody the content of the article? (25%) | 2/5 |
|  | Does the introduction give appropriate context? (25%) | 3/5 |
|  | Is the objective of the experiment clearly defined? (25%) | 3/5 |

## Analysis

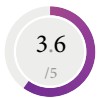

**3.6**
/5

Does the discussion adequately interpret the results presented? (40%)                    4/5

Is the conclusion consistent with the results and discussion? (40%)                    3/5

Are the limitations of the experiment as well as the contributions of the experiment clearly outlined? (20%)                    4/5