## [Reviewer Report · Review 1: Weather variability and COVID-19 transmission rates: a time series analysis based on effective reproductive number]

*Comments to the Author:* This study investigated the effects of air temperature and relative humidity on SARS-CoV-2 transmission in Wuhan, China. Great improvements have been made in this manuscript, such as the brief review of existing studies, the sensitivity analyses by testing different degrees of freedom, and more justification in the discussion part. There are several minor issues needed to be addressed before publication:

1. The authors used lag of 5 days for temperature and 8 days for RH in the statistical model. Explanations on how these lag durations were chosen is necessary. In addition, the selection of lag period could influence the final estimates. Sensitivity analyses using different lags are needed.

2. Line 94-96, it is good to perform these sensitivity analyses, but please display model statistics to support this statement.

3. In the description of the GAM, it is unclear which type of smoothing basis was used (i.e., bs=? in the s(.))?

4. There are still several typos in the text (e.g., line 49, “tete”), please check again in the revision.

## Score Card

### Presentation

4.0/5

Is the article written in clear and proper English?30%4/5

Is the data presented in the most useful manner?40%4/5

Does the paper cite relevant and related articles appropriately?30%4/5

### Context

4.5/5

Does the title suitably represent the article?25%4/5

Does the abstract correctly embody the content of the article?25%5/5

Does the introduction give appropriate context?25%4/5

Is the objective of the experiment clearly defined?25%5/5

### Analysis

4.4/5

Does the discussion adequately interpret the results presented?40%4/5

Is the conclusion consistent with the results and discussion?40%5/5

Are the limitations of the experiment as well as the contributions of the experiment clearly outlined?20%4/5

---

## [Reviewer Report · Review 2: Weather variability and COVID-19 transmission rates: a time series analysis based on effective reproductive number]

*Comments to the Author:* • The manuscript can be checked by native English speaker

• The main contribution and originality should be explained in more detail, is it the use of Bayesian Estimation? or the found associations?

• Discussion of related work in COVID-19 should be expanded with more recent work, this will better situate this paper in the context of the journal

• This paper has quite a few issues to check and perhaps to correct. But one major issue is to have and to support reproducibility of this work, to have comprehensive evaluation and applications carried out, from this work here.

• The wider interpretation of the results can be added for the better understanding of the analysis.

• A comparison with other methods should be explained with more detail to show the effectiveness of using the new approach== for example with the following two references:

1- Association between Weather Data and COVID-19 Pandemic Predicting Mortality Rate: Machine Learning Approaches, Zohair Malki, El-Sayed Atlama, Aboul Ella Hassanien, Guesh Dagnew, Mostafa A. Elhosseini and Ibrahim Gad, Journal of Chaos, Solitons &Fractals, Vol. 138,110137,2020.

2- ARIMA Models for Predicting the End of COVID-19 Pandemic and the Risk of a Second Rebound, Zohair Malki, El-Sayed Atlam, Ashraf Ewis, Guesh Dagnew, Ahmad Reda Alzighaibi, ELmarhomy Ghada, Mostaf A. Elhosseini, Aboul Ella Hassanien, Ibrahim Gad., May 27,2020, Journal of Neural Computing and Applications, May 272020-Accepted Oct.8th,2020.

oMany references are old and recent references should be appended for good comparison to related works.

## Score Card

### Presentation

3.0/5

Is the article written in clear and proper English?30%3/5

Is the data presented in the most useful manner?40%3/5

Does the paper cite relevant and related articles appropriately?30%3/5

### Context

2.8/5

Does the title suitably represent the article?25%3/5

Does the abstract correctly embody the content of the article?25%2/5

Does the introduction give appropriate context?25%3/5

Is the objective of the experiment clearly defined?25%3/5

### Analysis

3.6/5

Does the discussion adequately interpret the results presented?40%4/5

Is the conclusion consistent with the results and discussion?40%3/5

Are the limitations of the experiment as well as the contributions of the experiment clearly outlined?20%4/5